# Effect of the Genotypic Variation of an Aphid Host on the Endosymbiont Associations in Natural Host Populations

**DOI:** 10.3390/insects12030217

**Published:** 2021-03-04

**Authors:** Francisca Zepeda-Paulo, Blas Lavandero

**Affiliations:** Laboratorio de Control Biológico, Instituto de Ciencias Biológicas, Universidad de Talca, Talca 3460000, Chile; blavandero@utalca.cl

**Keywords:** endosymbionts, parasitoids, aphid, clone, host–endosymbiont dynamics

## Abstract

**Simple Summary:**

The host–endosymbiont complex could be a key determinant in spread and maintenance of the infection polymorphism of endosymbionts. Variation among host–endosymbiont complexes can contribute to genetic variation of a host species and then provide the necessary material for the operating coevolutionary dynamics. We studied the seasonal dynamic of facultative endosymbiont infections among different host clones of the grain aphid *Sitobion avenae* and whether their presence affects the total hymenopteran parasitism of aphid hosts at the field level. We observed that aphid infections in the field with endosymbionts increase over time, by favoring particular aphid clones closely associated with endosymbionts, but without an effect of endosymbionts on parasitism rate in the host populations. Our results highlight the importance of host–endosymbiont couples in shaping the prevalence and distributions of symbionts throughout nature and the success of their hosts as pests.

**Abstract:**

Understanding the role of facultative endosymbionts on the host’s ecology has been the main aim of the research in symbiont–host systems. However, current research on host–endosymbiont dynamics has failed to examine the genetic background of the hosts and its effect on host–endosymbiont associations in real populations. We have addressed the seasonal dynamic of facultative endosymbiont infections among different host clones of the grain aphid *Sitobion avenae*, on two cereal crops (wheat and oat) and whether their presence affects the total hymenopteran parasitism of aphid hosts at the field level. We present evidence of rapid seasonal shifts in the endosymbiont frequency, suggesting a positive selection of endosymbionts at the host-level (aphids) through an agricultural growing season, by two mechanisms; (1) an increase of aphid infections with endosymbionts over time, and (2) the seasonal replacement of host clones within natural populations by increasing the prevalence of aphid clones closely associated to endosymbionts. Our results highlight how genotypic variation of hosts can affect the endosymbiont prevalence in the field, being an important factor for understanding the magnitude and direction of the adaptive and/or maladaptive responses of hosts to the environment.

## 1. Introduction

Facultative bacterial endosymbionts are ubiquitous in many insects as they may produce ecologically important effects in their insect hosts, contributing to the host’s adaptation to their environment [1]. Some beneficial traits conferred by facultative endosymbionts (i.e., not essential for the host insect’s survival) involve protection against natural enemies [2,3,4,5,6], providing resistance to heat stress [7] or influencing host-plant use [8,9,10]. A well-studied adaptative trait mediated by facultative endosymbionts is the defense again parasitoid wasps (especially hymenopteran parasitoids), which have been reported in two groups of insect hosts including Drosophila flies [11,12,13] and aphids [2]. Some facultative endosymbionts can directly improve host survival by negatively affecting the development and survival of immature stages of parasitoid wasps through toxin-based protection [14]. Thereby, defensive endosymbionts can mediate the parasitoid–insect interactions by protecting their insect hosts from parasitism pressures in the field. Despite the benefits of carrying endosymbionts, these are frequently found in intermediate frequencies in nature, ranging from uncommon (10% or lower) to prevalent (>80%) in host populations [15,16]. Field data have shown that endosymbiont prevalence within natural host populations could be highly dynamic over time and space [17], contrasting from unidirectional changes observed in populational cage experiments, where the beneficial endosymbiont frequency has been observed to increase over time under selective pressures [18]. Fluctuating selective pressures in the field could generate endosymbiont instability if they represent a cost to their insect hosts under determined environmental conditions [15]. It has been suggested that endosymbionts could be maintained in host populations by balancing selection, due to temporal changes of the benefits and costs of endosymbiont harboring under field conditions [15].

However, current research on host–endosymbiont dynamics has failed to examine the genetic background of the hosts and its effect on host–endosymbiont associations in real populations [see 16]. Variation among endosymbiont–host complexes can contribute to genetic variation of host species (holobiont) and then provide the necessary material for the operating coevolutionary dynamics [19]. Host–endosymbiont partners can shape an adaptive complex on which selective pressures of the environment can act, being maintained, and spread in the next generations [20]. Natural selection acting on the variation among host–endosymbiont complexes could have an important role maintaining and spreading the infection polymorphism of facultative endosymbionts on insect host populations [15]. Indeed, previous studies have shown evidence of the specificity of the associations between endosymbionts and their host lineages in natural populations of aphid species [21,22].

Aphid species are a major insect crop pest, and they represent a model system in the study of the host–endosymbiont interactions. Aphids (Homoptera: Aphididae) present reproduction by parthenogenesis, which could maintain clonal lineages harboring bacterial endosymbionts over time, through maternal symbiont transmission [1]. The grain aphid *S. avenae* is a worldwide pest of economically important cereal crops, also found in pasture grasses. Like other aphid species, the grain aphid has showed common infections with facultative endosymbionts in their natural populations [22,23,24,25,26,27]. Some of the commonly studied aphid endosymbionts include *Hamiltonella defensa, Regiella insecticola, Serratia symbiotica, Rickettsia, Spiroplasma* and X-type [12,13]. For instance, the defensive role of facultative endosymbionts has been well-studied in the common aphid endosymbiont *Hamiltonella defensa*, which can confer protection against several parasitoid species at least five aphid species [14]. *H. defensa* is well known for conferring a strong resistance against the parasitoid *Aphidius ervi* in the well-studied pea aphid *Acyrthosiphon pisum* [2,3] as well as resistance against the parasitoids *Lysiphlebus fabarum* and *Aphidius colemani* in the black bean aphid Aphis fabae [28,29] and to the parasitoids Binodoxys communis and *B. koreanus*, but do not confer protection against *Aphidius colemani* nor *Lysiphlebus orientalis* in the cowpea aphid, *Aphis craccivora* [30]. However, evidence of the role of facultative endosymbionts in the grain aphid *S. avenae* is still contradictory. On the one hand, laboratory and field-based studies suggest no protection-mediated role of common facultative endosymbionts (*H. defensa* and *R. insecticola*) against main parasitoid species of *S. avenae* [22,23]. While other field studies have shown a potential protective effect of endosymbionts in *S. avenae* against an assemblage of parasitoids [26]. As well as, increasing evidence for host-plant use mediated effects of endosymbionts have been reported for *S. avenae*, which show a significant variation among host clones on different host plants [31,32,33]. Here, we have addressed the seasonal dynamics of endosymbiont infections among different host clones of the grain aphid (*S. avenae*) on two cereal crops (wheat and oat) and whether their presence affects the total hymenopteran parasitism of aphid hosts at the field level.

## 2. Materials and Methods

### 2.1. Aphid Sampling and DNA Extraction

Live aphids were collected on four replicated oat (*Avena sativa*) and wheat (*Triticum aestivum*) fields during one growing season in Chile (autumn sowing to summer harvest). Pairs of oat and wheat fields were sampled in four different localities (eight cereal fields in total) at a 15-day interval, starting at the end of September to late December 2014 (seven sampling dates). The distance between localities ranged between 4 and 20 km. Live aphids were collected by sampling 100 tillers separately within each field. In order to minimize resampling the same aphid genotypes, each sampled tiller was separated by at least 20 m. In relation to the pest management of the crop fields studied, we have knowledge of insecticide applications in the midseason in some wheat fields studied. Even so, the aphid abundance was major on this cereal host in comparison to oat (see in results). Indeed, this type of pest control is less important in the cereal crops, due to the highly efficient biological control performed by introduced parasitoid species on cereal aphid pests in Chile [34].

Aphid density was estimated by counting the total number of aphids present on these 100 tillers from each cereal field during the different sampling dates. DNA extraction was performed in a subset of 50 aphid individuals were randomly chosen from each cereal field and sampling date to be used in the genetic analysis (DNA extraction). In the fields where the sampled number of aphids was less than 50, the totality of the collected specimens was examined. DNA extraction was individually performed for each aphid specimen using the “salting out” method described by Sunnucks and Hales [35]. The quantification and quality of the extracted DNA was examined by absorbance using Infinite 200 PRO NanoQuant (TECAN) and by electrophoresis in 0.8% agarose gels. Each individual DNA extraction was normalized to a concentration of 5 ng/μL and kept at −20 °C until later PCR analysis.

### 2.2. Microsatellite Genotyping of Aphid Individuals

The characterization of the microsatellites genotypes of aphids was conducted by amplifying eight microsatellite loci (S3.43, S5.L, S16b, S30, Sa4Σ, Sm11, Sm10 and Sm17) described for the grain aphid *S. avenae* [36,37]. Microsatellite genotyping was performed by capillary electrophoresis (Macrogen, Seoul, Korea) and the multilocus genotypes were analyzed using the (Softgenetics, State College, PA, USA). PCR reactions were performed in a volume of 10 μL, including 1 μL of buffer 10×, 0.2 mm dNTP, 2.5 mm MgCl2, 0.05 μL Taq (5U/μL), 0.25 μm of each forward, reverse and -M13 universal primer and 1.3 μL DNA. The PCR conditions consisted of 94 °C for 2 min, followed by 35 cycles of 94 °C for 40 s, annealing primer specific for 40 s and 72 °C for 45 s and final extension of 72 °C for 2 min.

### 2.3. Molecular Screening of Bacterial Endosymbionts and Parasitism in Aphid Individuals

The molecular identification of endosymbiont infections and parasitism status was individually performed in each aphid. A previous screening of bacterial community diversity using 16S rRNA amplicon sequencing revealed a low diversity of facultative endosymbionts in the grain aphid *S. avenae*, with only two endosymbiont species: *R. insecticola* and *H. defensa* [38]. In the present study, the detection of facultative endosymbionts in each aphid individual was performed by the PCR diagnostic of a region of bacterial 16S rDNA, which included common facultative endosymbionts previously detected in aphid species. A multiplex PCR for each aphid DNA was performed, using specific primers of five common facultative endosymbionts of aphids previously reported [39]. Which included *H. defensa* (PABS480R), *R. insecticola* (PAUS16SR), *Serratia symbiotica* (PASS1140R), *Spiroplasma* spp. (Spi1500R) and *Rickettsia* sp. (Ric600R) together with a universal forward primer (16SA1). In addition, the obligate aphid endosymbiont *Buchnera aphidicola* (Buch270R) was used as positive control of the reaction, following the PCR condition used by Zepeda-Paulo et al. [22]. The different facultative endosymbionts were discriminated according to the size of the amplicons (bp) visualized in 1.5% agarose gels stained with Redgel (Biotium, Hayward, CA, USA) using GeneRuler 100 bp plus a DNA ladder (Thermo Scientific, Waltham, MA, USA) as previously described by Peccoud et al. [39].

A molecular approach for assessing parasitism of the aphid parasitoid-group was used by amplifying a specific region of the 16S gene of the DNA of parasitoid eggs or larvae parasitizing aphids using a primer pair (AphG-S458: WATAATYTTAAGTCWAATCTGCC and ParG-A462: AARTTCTAWAGGGTCTTMTCGTCT) designed to amplify a complete set of all primary aphid parasitoids within the Aphidiinae according to Ye et. al. [40]. In previous lab and field studies, these markers proved to correctly determine the total parasitism rates for the present species of Aphidiinae in Chile [22,41]. Reactions were performed in a volume of 10 μL, including 1 μL of buffer 10×, 0.2 mm dNTP, 3 mm MgCl2, 0.1 μL Taq (5U/μL), 1 μm of each primer and 1 μL DNA. The PCR conditions consisted of 94 °C for 3 min, followed by 35 cycles of 94 °C for 30 s, 62 °C for 30 s and 72 °C for 1 min and final extension of 72 °C for 5 min. Positive controls were used by amplifying known parasitoid DNA in each PCR reaction. The presence of aphid parasitism was assigned by visualizing PCR amplicons and controls in 1.5% agarose gels.

### 2.4. Data Analysis

The number of aphids per 100 tillers was analyzed using generalized linear mixed models (GLMMs) with a Poisson distribution in the lme4 package in R version 2.14.2 [42,43]. The proportion of different aphid clones was studied using a data set that included the four most prevalent aphid genotypes, which represented about 84% of the total sample. The number of aphids and variation in the proportion of aphid clones were studied among the fixed factors include sampling date (seven sampling dates), cereal host (oat and wheat) and localities, while the origin field of the sampled aphids and a temporal structure were assigned as nested random effects (fields within date) into all GLMMs. The proportion of parasitized aphids and the proportion of infected aphids by facultative endosymbionts were studied using GLMMs with a binomial distribution [43]. The fixed factors included aphid genotype (aphid clone), cereal host, sampling date and endosymbiont status for the variable parasitism rate. All GLMMs included the origin (four oat and wheat fields) of the sampled aphids, and a temporal structure as nested random effects (field within date) into all models. Model simplification from saturated models was performed using the *anova* function of the car package [44]. Multiple comparisons among the levels of factors were performed using the Multcomp package [45]. Randomness of the residuals and overdispersion were checked and corrected for all GLMMs according to Bolker et al. [46].

## 3. Results

### 3.1. Aphid Density and Prevalence of Aphid Clones

A total of 3624 *S. avenae* aphids were collected on oat and wheat fields across the growing season (Appendix A). Significant differences in the number of aphids across sampling dates (X2 = 17.97; *p* < 0.0001), host-plants (X2 = 4.13; *p* = 0.041), and interaction of both factors (X2 = 4.55; *p* = 0.032) were found, but nonsignificant differences among localities were observed. A significant increase in the mean number of aphids was observed across sampling dates (Figure 1), observing a significantly larger mean number of aphids on wheat (80.66 ± 14.44 SE) than on oat (38.85 ± 4.42 SE) for the whole sample. The frequency of aphids begins to increase significantly on wheat from the middle of the season (fourth date) in comparison to oats where the frequency of aphids was observed constant across the growing season (Figure 1).

A total of 16 aphid clones based on their multilocus genotypes were identified from the subset data analyzed (1651 aphids) (Appendix A). Four aphid clones were found at a high prevalence in the crop fields (G1, G2, G3 and G4), accounting for about 84% of the whole sample analyzed from host plants across the growing season (Appendix A). Significant differences in the mean proportion of the four most common clones of aphids were observed from the whole sample (X2 = 47.79; *p* < 0.0001). The aphid clone G1 was significantly more prevalent (at a proportion of 0.40 ± 0.01 SE) than the aphid clones G2 (0.27 ± 0.01 SE), G3 (0.24 ± 0.01 SE) and G4 (0.14 ± 0.01 SE) across the whole sample in both host-plants studied. Predominance of aphid clones were observed to vary through the growing season (host genotype × sampling date interaction) (X2 = 48.88; *p* < 0.0001) (Figure 2), due to the dominance of the aphid clone G1 during the first three dates of the season in comparison to the other clones, which did not vary between host plants (Appendix A) or different localities (Appendix A). However, the prevalence of aphid clone G1 significantly decreased from mid-season, being absent on the last date studied, which was true in both host-plants (Figure 2). In contrast, the other clones significantly increased their predominance across the season, like the aphid clone G2 or the aphid clone G4 (which was not observed in the first sampling date) (Figure 2). Finally, three of the four common clones (G2, G3 and G4) were observed in a similar proportion at the end of the season (Figure 2).

### 3.2. Endosymbiont Infections and Parasitism in the Aphid Clones

Of a total of 1619 aphids analyzed for endosymbiont infections (subset data in materials and methods); 1068 aphids showed single infections (only one species of facultative endosymbiont by each individual) with the endosymbiont *R. insecticola*, 182 aphids with *H. defensa*, and only one individual showed a single infection with the endosymbiont *Spiroplasma* spp., while 362 corresponded to uninfected aphids (no endosymbionts detected). In addition, only six individuals showed double infections with the endosymbionts *R. insecticola* and *H. defensa.* Due to the low number of coinfected aphids, only the proportion of single infections with *R. insecticola* and *H. defensa* among the four common aphid clones across the season and between different host-plants were analyzed. The mean proportion of infection by *H. defensa* varied significantly between host-plants (X2 = 7.47; df = 1; *p* < 0.0001) and across sampling dates (X2 = 4.80; *p* = 0.028), a higher mean proportion of infected aphids with *H. defensa* in oat (0.21 ± 0.02 SE) in comparison to wheat plants (0.08 ± 0.01 SE), with an decrease in infection by *H. defensa* across the growing season for both host-plants (Figure 3), without any significant interactions (aphid genotype × sampling date, host-plant × sampling date and aphid genotype × host-plant). Infections with *R. insecticola* between host plants and the host-plant × sampling date interaction were not significantly different (Figure 3).

On the other hand, the mean proportion of infected aphids with *R. insecticola* (X2 = 36.96; df = 3; *p* < 0.0001) and *H. defensa* (X2 = 56.4; df = 3; *p* < 0.0001) varied significantly among the common aphid clones studied. The infection with *R. insecticola* was similar for three aphid clones G2, G3 and G4 with a mean proportion of infection of 0.86 (±0.02 SE) for each of them, compared to the aphid clone G1, which showed a significantly lower mean proportion of infected aphids (0.19 ± 0.02 SE). In contrast, the mean proportion of aphids infected with *H. defensa* in the aphid clone G1 (0.35 ± 0.02 SE) was greater in comparison with the other aphid clones G2 (0.03 ± 0.009 SE), G3 (0.01 ± 0.006 SE) and G4 (0.02 ± 0.011 SE). Significant differences in the proportion of infected aphid clones by *R. insecticola* across the growing season (aphid genotype × sampling date interaction) were observed (X2 = 22.88; *p* < 0.0001) (Figure 4); significantly lower infection levels with *R. insecticola* were recorded during the first part of the season for the aphid clones G1 (first four dates) and G4 (first three dates), in comparison to the high levels of infection observed for clones G3 and G2 in all sampling dates and irrespectively of the host plant (Appendix A).

The parasitism rate was studied among the most common aphid clones, considering their endosymbiont infection status (infected with Hamiltonella or Regiella and uninfected) and the different two host-plants at the different sampling dates and localities. The parasitism rates vary significantly between aphids infected by *H. defensa* (0.40 ± 0.04 SE), *R. insecticola* (0.52 ± 0.02 SE) and uninfected aphids (0.48 ± 0.03 SE), but did not vary among the aphid clones: G1 (0.41 ± 0.03 SE), G2 (0.58 ± 0.02 SE), G3 (0.49 ± 0.03 SE) and G4 (0.52 ± 0.04 SE), neither for the interaction between aphid genotype × infection status nor among localities studied. However, a significant effect of the host-plants (X2 = 29.22; *p* =0.002) across the whole season was observed, the mean proportion of parasitized aphids was significantly higher on oat (0.54 ± 0.02 SE) than wheat (0.47 ± 0.02 SE). A significant effect of the sampling date (X2 = 75.60; df = 39; *p* < 0.001), due to an increase in the mean parasitism rate was observed; during the first date no parasitized aphids were detected, with a maximum parasitism rate during the fifth sampling date (mean proportion of 0.7 ± 0.03 SE) (Figure 5). The host-plant × sampling date interaction showed a significant effect (X2 = 13.19; df = 6; *p* =0.041) on the parasitism rate, due to a lower parasitism of the first sampling date on oat, which increased significantly towards the middle of the season in comparison to what was observed for wheat (Figure 5).

## 4. Discussion

When observing the seasonal host genotype-endosymbiont dynamics in the populations of *S. avenae* on both host-plants, we first observed a high prevalence of bacterial facultative endosymbionts among aphid clones within both wheat and oat. A large fraction of the aphid populations here studied was constituted by a few predominant aphid clones (G1, G2, G3 and G4 clones) that represented above 84% of the overall sample. *R. insecticola* was the predominant facultative endosymbiont species (~66%) in comparison to *H. defensa*, which was observed at a lower frequency in the studied populations (~22%). Superinfections (i.e., multiple endosymbionts in an aphid individual) were rare in the host populations studied, such situation has been previously reported in other aphid species and has been associated to costs of multiple infections on aphid fitness [see 16].

Variable infection levels with the facultative endosymbiont *R. insecticola* have been previously reported for *S. avenae* populations in different regions, including Morocco (75.6%), England (34%), China (5% and 15%); Germany (~40%), as well as in Chile (54%) [22,23,24,25,26,27].

Interestingly, we observed that irrespective of the host-plant studied here, there are rapid seasonal shifts in the frequency of the facultative endosymbionts in the natural host populations studied. Unlike *H. defensa*, the frequency of the more prevalent facultative endosymbiont, *R. insecticola*, increased most probably through the population growth of their hosts, achieving high levels of infection on both oats (~93%) and wheat (~87%) towards the end of the studied season.

Our findings highlight that the aphid clones strongly influence the endosymbiont prevalence in the natural populations., aphid clones harboring an endosymbiont species were positively selected. Our results showed major differences found in the percentages of infected aphids when compared to uninfected aphids for the studied common aphid clones, suggesting that the host genotype determines the endosymbiont associations found at the field level. For example, two of the four more common aphid clones (G2 and G3) showed high infection levels with *R. insecticola* (>84%), irrespectively of the host-plant and sampling date, while the other common aphid clone (G1 or G4) exhibited a lower proportion of Regiella-infected aphids at the beginning of the season, significantly increasing their endosymbiont infection levels as the season carried on. A previous survey of these Chilean aphid populations had already provided evidence of the high specificity in the host genotype-endosymbiont associations for several common aphid clones sampled in a previous season [22]. Therefore, suggesting that associations between aphid clones and endosymbionts can be held over time (or at least during consecutive years). Aphids are mostly apomictic parthenogenetic, and under this form of clonal reproduction (with high rates of population increase), the maternal transmission of bacterial endosymbionts through telescopic generations, may favor the persistence of associations between host lineages and their endosymbionts [47,48,49]. Associations between host and endosymbiont lineages have been found in the natural host populations, which are thought to be mainly shaped by the environment [21]. As we and previous studies have suggested, highly persistent host clone-endosymbiont associations could be maintained within host populations by inter-clonal selection in temporally variable environments. For instance, widespread and time-persistent aphid clonal lineages (sometimes referred to as “superclones”) have been shown to present a high ecological success in agroecosystems [50,51,52,53], perhaps favored as well by their associated endosymbionts.

Aphid host–endosymbiont couples could be favored by natural selection when they provide net fitness benefits to their insect hosts and selective pressures are prevalent in the environment [15]. Therefore, identifying the selective pressures that affect particularly successful host genotype-endosymbiont associations will be essential for understanding the role of endosymbionts in the host adaptation to their environment, as well as unveiling the mechanisms behind the success of aphid pests. However, individual-level and laboratory studies will fail to grasp the functional role of host endosymbiont interactions at the populational and community level, therefore only by inspecting the dynamics in real world populations can these complex interactions be understood. In this respect, individual level studies have suggested that *R. insecticola* can have a functional role in the host-plant use by their hosts [54,55], as endosymbiont harboring improves the performance of pea aphids on white clover (Trifolium repens) [8]. For the grain aphid *S. avenae*, evidence for some effect of *R. insecticola* on the host-plant use has been reported. For instance, Wang et al. [31] studied the effect of *R. insecticola* on the fitness (i.e., developmental time and fecundity) of seven naturally infected *S. avenae* clones (and cured of endosymbionts using antibiotic treatment) on wheat, oat, and rye, finding that *R. insecticola* harboring negatively affects the performance of aphids on rye, while showing a neutral effect on both wheat and oat. Similarly, Łukasik et al. [56] reported no fecundity costs or benefits associated with *R. insecticola* harboring in *S. avenae* aphids. In contrast, Luo et al. [57] found that *R. insecticola* endosymbiont would negatively affect several life history traits of their aphid hosts on wheat for two aphid genotypes of *S. avenae*, suggesting that *R. insecticola* could decelerate the population growth of these aphid clones in the field. Nevertheless, a detrimental effect of *R. insecticola* on the fitness of *S. avenae* clones, is not consistent with our field data, where higher levels of infection with *R. insecticola* were observed in the different aphid clones studied. These results are consistent with a previous experimental observation for a *S. avenae* clone; where naturally Regiella-infected aphids presented a higher population growth on wheat than on barley, while harboring *R. insecticola* negatively affect the growth of aphids on barley when compared with naturally uninfected aphids [33]. Notably, this experiment was carried out using a single aphid clone collected in Chile [G2 in 22], which correspond to the same multilocus genotype of the aphid clone G1 observed predominantly during the period of crop colonization in this present study. This aphid clone (G1) showed a clear upward trend in the infection levels with *R. insecticola* on wheat and oat. However, unlike the other common clones, their prevalence in the field decreased significantly across the growing season (G1 was not found on the last date for both cereal hosts). This observation could be explained by the effect of trade-offs in traits related to this aphid clone’s fitness associated with the presence of this endosymbiont [58]. For example, endosymbionts may pose costs by increasing some reproductive traits at the expense of other correlated traits (e.g., number of nymphs, fecundity, body size at maturity, development time and per capita growth rate) and/or other physiological traits associated with host-plant use (e.g., detoxification of plant chemical defenses). Regardless, costs or benefits of endosymbionts could vary among host lineages, which could explain the major differences in the infection levels and prevalence observed among different aphid clones through the growing season. Increasingly, experimental evidence shows that host–endosymbiont interactions could be much more complex than previously thought. For instance, the effect of facultative endosymbionts on their aphid hosts can considerably vary among aphid clones, suggesting that the genetic background of insect hosts could strongly influence the effect of endosymbionts [6,10]. Also, variation in the strength of effect provided by endosymbionts in their hosts have been reported among related endosymbiont strains [3,56,59], supporting the idea that genotype × genotype interactions between hosts and facultative endosymbionts are a key factor for understanding the role of endosymbionts on its host ecology and host–endosymbiont dynamics in nature. In this respect, it should be noted that several laboratory-based studies set up infected clones by transinfection of endosymbionts from donor clones to other clones of aphids, which could result in phenotypic modifications due to maladaptation of the new host–endosymbiont associations [60,61,62]. Besides, there is evidence of endosymbiont strain fidelity, as the ability of endosymbionts to establish in hosts and then be transmitted to offspring can be affected by the endosymbiont’s genetic similarity, probably due to co-evolutionary processes between the host’s immune system and the bacterial symbionts [56]. Phenotypic variation among host clone-endosymbiont strain associations on which host-level natural selection may act, could shape coadapted complexes in the natural populations [19]. However, if specific host–endosymbiont interactions will entail at least a maintenance cost, they could be lost again due to selection on the hosts, if they do not confer any fitness benefit in the specific environment where they exit [18,63]. Thereby, operating co-evolutionary dynamics between hosts and endosymbionts could enable the establishing and maintenance of the polymorphism occurring in the natural host populations [15]. A greater understanding of the coevolutionary processes between host lineages and endosymbiont lineages is particularly important in pest systems, as endosymbiont carrying can affect traits that contribute to a rapid evolution of the invasiveness of host pests. In aphid species, selection favoring coadapted host–endosymbiont complexes could allow the maintenance and spread of some aphid clones over others in host pest populations in the field, becoming a key determinant to explain the adaptive potential and their pest status [64]. This situation has been reported in other crop pests, such as the whitefly *Bemisia tabaci*, where specific associations between symbiotypes and pest host biotypes were found in host populations, endosymbionts playing a key role on the spread of invasive host lineages [65,66].

### Relationship among Endosymbiont Infections and Parasitism Rates in the Field

Natural enemies can be a strong selective pressure for facultative endosymbionts as well as their hosts in nature, as a range of them confer protection to their insect hosts against parasitoids, fungal pathogens, and predators. Notwithstanding, the results here presented show a lack of evidence for an effect of the facultative endosymbionts, *H. defensa* nor *R. insecticola* on the overall parasitism rate in *S. avenae* clones in the field. There are no significant differences in the parasitism rates among infected aphid clones with facultative endosymbionts (Hamiltonella nor Regiella) and uninfected in the overall season on both oat and wheat fields. In agreement with our results, Ye et al. [40] reported that in the grain aphid *S. avenae* facultative endosymbionts did not affect parasitoid oviposition behavior in the field. Although, the molecular detection of parasitoid eggs and larvae parasitizing aphids represent only a proxy of the mortality produced by female parasitoids. Molecular detection does not allow one to detect whether parasitized aphids will mummify (i.e., parasitoid pupa) as the detection of an egg inside an aphid does not guarantee the development of a parasitoid and aphid mortality. In this sense, they reported that *S. avenae* mummies collected in the field exhibited a lower endosymbiont infection than living parasitized aphids, suggesting that endosymbionts could affect overall parasitoid survival in the host. The relationship between facultative endosymbiont prevalence and parasitism also needs to be interpreted cautiously as aphids are attacked by a diverse guild of parasitoids in the field [41,67,68], but facultative endosymbionts have shown a defensive effect for a limited range of parasitoid and aphid species and even the strength of protection can differ among facultative endosymbiont strains and species [15,16]. In effect, different laboratory-based studies have shown that *H. defensa* does not affect the development of two parasitoid species, *A. ervi* and *Ephedrus gladiator* in the grain aphid *S. avenae* [22,23]. Although, Łukasik et al. [23] evidenced that *A. ervi* female parasitoids preferentially oviposit on Hamiltonella-free aphids when given a choice between infected and uninfected insects. Such oviposition preference would favor *H. defensa*-infected aphids in the natural populations, however our results do not support this latter situation, as we evidenced a very low prevalence of Hamiltonella in the aphid populations, which was mainly influenced by one single aphid clone. In contrast, the proportion of aphids harboring *R. insecticola* increased in a similar sense to the total parasitism rate, what could be a signal of positive selection for endosymbionts related to parasitism pressures. However, two experimental studies have already reported that *R. insecticola* does not confer protection against the parasitoid *A. ervi* in the aphid *S. avenae* [22,56] which is the most successful parasitoid species within a parasitoid assemblage (mainly composed by three parasitoid species) attacking Chilean populations of *S. avenae*, becoming a dominant species in this system particularly in periods of high aphid density [68].

Similarly, in other aphid species like the pea aphid, *R. insecticola* does not confer resistance against parasitoids [2,69] and only one strain of this endosymbiont has been shown to protection against the parasitoid *A. colemani* in the aphid *Myzus persicae* [70]; thus, representing a limited evidence for a defensive role against parasitoids for *R. insecticola*. Even more Regiella-infected *S. avenae* could increase the risk of parasitism [57] and even predation [33]. A recent behavioral study found that *R. insecticola*-harbored aphids were more predated by the coccinellid predator Hippodamia variegata than uninfected aphids of the grain aphid *S. avenae* [33]. *H. variegata* is the most abundant species (~50% of all coccinellid individuals) among the coccinellid group attacking *S. avenae* in the field [67]. Thus, it would be expected that an increased susceptibility of Regiella-harboring *S. avenae* to predation should negatively affect the frequency of *R. insecticola* within host populations. However, this situation is opposite to what is observed in our study. Taking all the above exposed into account, findings supporting a defensive role for *R. insecticola* against natural enemies like parasitoids or predators are scarce, and even harboring Regiella showed increased attack risk by natural enemies. However, the rapid increase in frequency of *R. insecticola* in the field, as reported here, suggests that additional selective pressures should be operating. Despite this, *R. insecticola* could mediate the defense against the fungal pathogen *Pandora neoaphidis* [4], potentially being beneficial for *R. insecticola*-harbored aphids at the field level. Additionally, variations in the strength of protection against the fungal pathogen *P. neoaphidis* have also been observed among different endosymbiont strains of *R. insecticola* studied in two *S. avenae* clones [56]. Other endosymbiont-related functions have been recently suggested for heat stress by *R. insecticola* in the pea aphid [71], but such a protective effect differs from previous studies [59,72], suggesting that effects mediated by facultative endosymbionts can be dependent on symbiont and host genotypes or their interaction [71].

## 5. Conclusions

Accordingly, several factors, such as benefits or costs of infection, symbiont transmission rates, as well temporal and spatial variations of selective pressures are expected to affect the endosymbiont dynamics in the environment. Here, the study of the temporal dynamics of host–clone-endosymbiont interactions reveals an important role of host–endosymbiont couples in shaping the prevalence and distributions of symbionts throughout nature. We observed rapid seasonal shifts in the frequency of facultative endosymbionts, which were related to specific host–endosymbiont associations, suggesting that endosymbionts are important for the host’s adaptation to its environment and then on the invasive potential of insect hosts. Therefore, specific clone-endosymbiont associations are being favored, and possibly increasing their chances for success as agricultural pests. Despite multiple evidence coming from experimental and field data for the importance of the genotype × genotype interactions for the effect of endosymbionts on their insect hosts, these have remained largely unexplored in studies on endosymbiont–host dynamics. Further studies on the effect of endosymbionts on their aphid hosts, as a protective effect against natural enemies, require exploring especially significant aphid-endosymbiont combinations with different host backgrounds coming from the field. The intraspecific variation in the host–endosymbiont interactions can be key for knowledge of the magnitude and direction of the adaptive and/or maladaptive responses of hosts to the environment and for understanding the role of facultative endosymbionts on the success of their hosts as pests.

## Figures and Tables

**Figure 1 insects-12-00217-f001:**
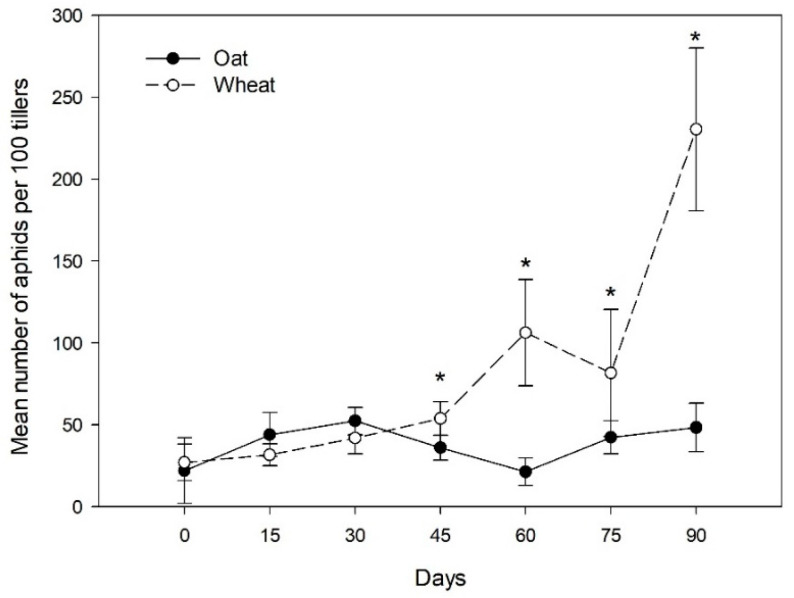
Aphid abundance on different host plants across the season. Mean number of aphids (±SE) on oat and wheat fields in the different sampling dates across one season. * Represents significant differences in the aphid abundance between host plants.

**Figure 2 insects-12-00217-f002:**
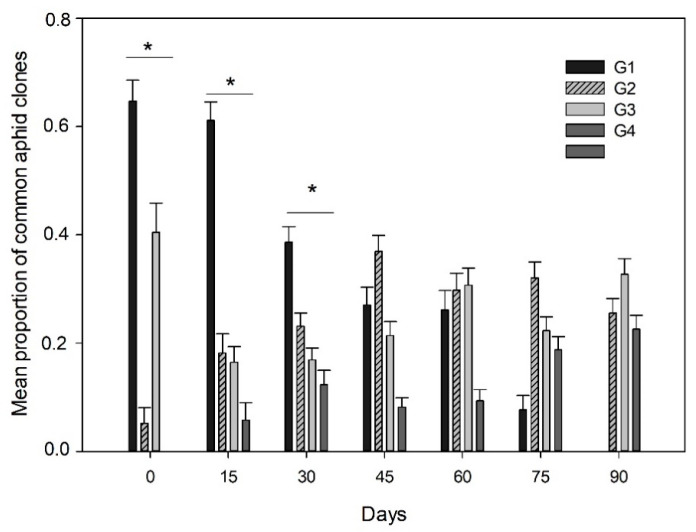
Predominance of common aphid clones across the season. Mean proportion (±SE) of the aphid clones G1, G2, G3 and G4 in the different sampling dates across one season. * Represents significant differences in the mean proportion of aphid clones in a sampling date.

**Figure 3 insects-12-00217-f003:**
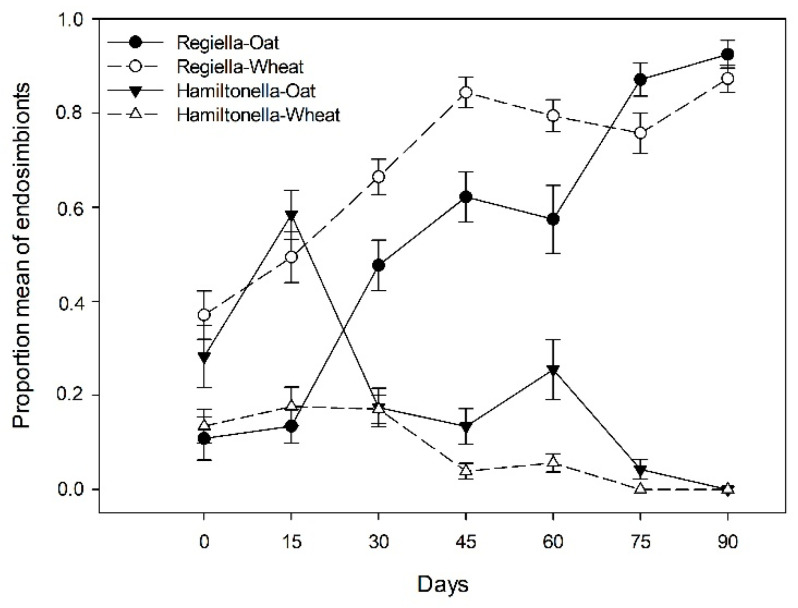
Mean proportion of infected aphids with facultative endosymbionts *R. insecticola* and *H. defensa* on wheat and oat across one season.

**Figure 4 insects-12-00217-f004:**
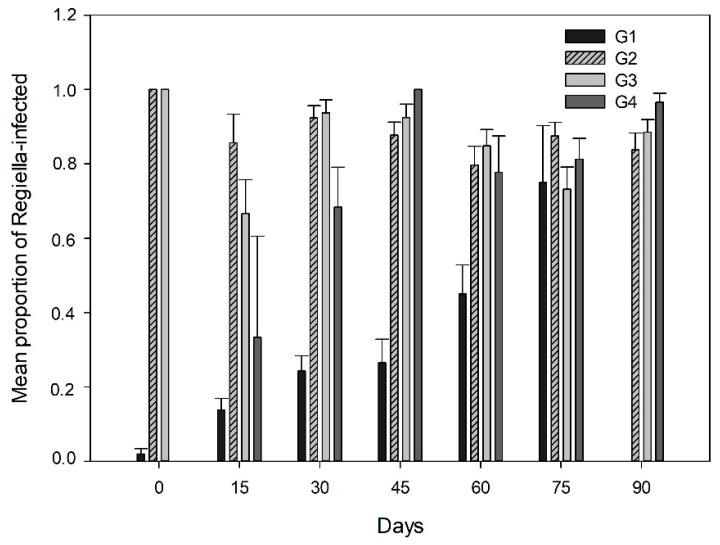
Mean proportion (±SE) of infected aphids with the common endosymbiont *R. insecticola* in the different aphid clones studied (G1, G2, G3 and G4) across one season.

**Figure 5 insects-12-00217-f005:**
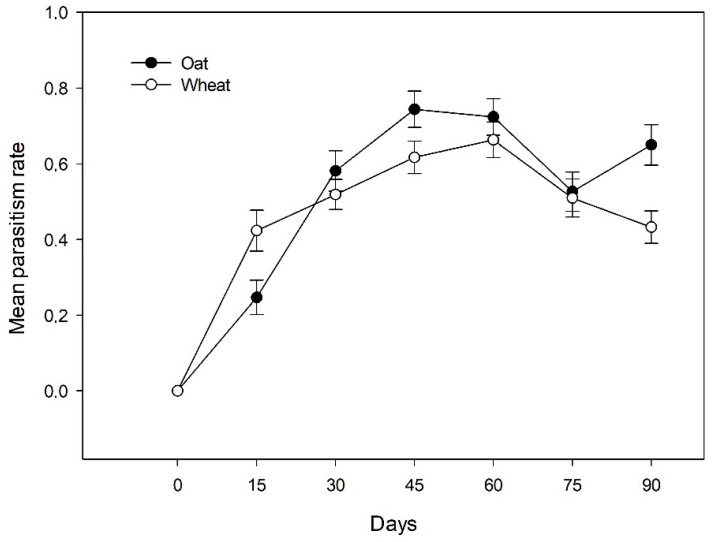
Mean parasitism rates in oat and wheat fields across the different sampling dates.

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
