# Peer review of "Effect of the Genotypic Variation of an Aphid Host on the Endosymbiont Associations in Natural Host Populations"

_insects, 2021, doi:10.3390/insects12030217_

Round 1
Reviewer 1 Report
In this study, the authors address the seasonal dynamic of endosymbiont prevalence in the grain aphid Sitobion avenae on two cereal crops.
In general, this manuscript is well written. Further, the survey and the analyses are well conducted. I particularly appreciate the investigation of the parasitism rate as a selective pressure. This study provides an original description of the dynamic of host-endosymbiont prevalence and highlights complex temporal patterns. Unfortunately, this study does not identify selective pressures to explain this complexity but discusses several axes of investigation. Due to the quality of the study, I have only minor comments.
Minor comments
Abstract: Please remove lines 20 to 27
Introduction and Discussion: I suggest to organize the introduction and the discussion parts in different paragraphs to facilitate the reading.
Line 97: The authors investigated four different localities but this factor is never exposed in the results nor in the discussion. Are there any differences between localities?
Lines 207-208: The numbers of coinfected individuals are rare. Is it possible to test if coinfections are less frequent than expected under random assortment?
Figure 1: Please specify the meaning of the stars.
Figures 2 et 4: Please use the same colors for the different genotypes in the two figures (colors for G3 and G4 are inverted).
Reviewer 2 Report
The authors tested the seasonal dynamics of endosymbiont prevalence and parasitoid infections in aphids in two different crop systems. They collected aphids from four pairs of oat and wheat fields over the course of a growing season and tested for the presence of facultative endosymbionts and parasitoids. They found significant interactions between endosymbiont infections over time, among different aphid genotypes, and between different crops. They found no interaction between parasitoid infection and endoysmbionts, but there were significant interactions between parasitoid infection and sampling date and host plant. These results highlight the interaction between host genotype and variation in endosymbionts in different temporal and ecological backgrounds.
I think this is a nice study that presents some interesting results. Your Methods and Results sections are well organized and seem to be thorough. You also nicely connect your results to previous work in this field. My only major comments relate to a lack of some details in the Introduction and questions about the field methods:
1. The role of hymenopteran parasitism needs to be more developed in the introduction. Currently, you only briefly mention some information about parasitoids towards the end of the introduction, but given how much you devote to testing for parasitoids in the study, I think you should devote more space to it here in the intro. It probably deserves its own paragraph. What do we know more generally about endosymbiont protection against parasitoids? How widespread is this in insects? I think adding information like this will better introduce your system and questions.
2. Also related to parasitoids, what about adults of the hymenopterans? What is the basic life history of these insects? Do they emerge at certain times of year? Could this explain any of the patterns you find?
3. Were the pairs of oat and wheat fields separated by 4 to 20 km, or were all eight fields separated by these distances? Perhaps a diagram or map would be helpful, at least in the supplement. I think it could also be useful to include a supplementary table with the raw collection data for each field.
4. Were the fields treated with any pesticides or herbicides? Could this affect results?
Minor comments:
Line 14: Please use the full genus name here. Also, please write out the full genus name upon first use (e.g., Line 123)
Abstract: Remove the instructions at the beginning of the abstract. I’m assuming the Abstract actually begins on Line 27?
Italicize taxonomic names throughout
I’m having difficulty with saying endosymbionts “contribute” to host genetic variation, although perhaps this is the language of the field. I think this language would be more clear if you put it in the context of holobionts and hologenomes.
I think there should be a new paragraph beginning Line 70.
How did you choose the 50 aphid individuals from each field/date?
